# Actin-microtubule interplay coordinates spindle assembly in human oocytes

Johannes Roeles[1] & Georgios Tsiavaliaris[1]*

Mammalian oocytes assemble a bipolar acentriolar microtubule spindle to segregate chromosomes during asymmetric division. There is increasing evidence that actin in the spindle interior not only participates in spindle migration and positioning but also protects oocytes from chromosome segregation errors leading to aneuploidy. Here we show that actin is an integral component of the meiotic machinery that closely interacts with microtubules during all major events of human oocyte maturation from the time point of spindle assembly till polar body extrusion and metaphase arrest. With the aid of drugs selectively affecting cytoskeleton dynamics and transiently disturbing the integrity of the two cytoskeleton systems, we identify interdependent structural rearrangements indicative of a close communication between actin and microtubules as fundamental feature of human oocytes. Our data support a model of actin-microtubule interplay that is essential for bipolar spindle assembly and correct partitioning of the nuclear genome in human oocyte meiosis.

---

[1] Cellular Biophysics, Institute for Biophysical Chemistry, Hannover Medical School, Carl-Neuberg-Straße 1, 30625 Hannover, Germany. *email: tsiavaliaris. georgios@mh-hannover.de

Oocytes prepare for fertilization through meiotic maturation[1,2]. In mammals, the process is initiated with the breakdown of the nuclear envelope (NEBD) and the formation of a bipolar microtubule spindle in the center of the egg that undergoes a series of dynamic reconfigurations to capture, sort, and align the genetic material to the equatorial plate[3,4]. Under the active guidance of cytoplasmic actin, the spindle is then positioned to the cortex[5–9] causing the extrusion of one half of homologous chromosomes into the first polar body in a process known as asymmetric division[10]. Failures in this process, frequently leading to chromosome misdistributions, are known to affect embryo development causing infertility, miscarriages, and congenital diseases in humans[11,12].

A number of studies mainly conducted with the mouse oocyte model show that actin is not only crucial for spindle migration and anchorage[13], but it also protects the maturing oocyte from chromosome segregation errors by promoting the formation of kinetochore-fibers that mediate stable microtubule-chromosome attachments[14]. This mechanism is thought to reduce the frequency of erroneous chromosome movements that are responsible for misdistributions leading to aneuploidy. In human oocytes, meiosis is highly error-prone and segregation defects are observed more frequently than in mice or other organisms[15,16]. The increased risk for aneuploidy appears to be associated with the unstable nature of the meiotic spindle and aberrant chromosome attachments occurring during spindle assembly[17]. These critical features of meiosis have hardly been studied in human oocytes; particularly the role of actin within the spindle during oocyte maturation remains poorly understood.

Here, by analyzing a total of more than 500 human oocytes donated from 184 women using multi-color three-dimensional (3D) confocal fluorescence microscopy, we investigated the spatiotemporal organization of actin and microtubules at the meiotic spindle of human oocytes simultaneously. Our further analysis identified that the meiotic actin spindle is composed of tropomyosin-decorated β- and γ-actin fibers that closely cooperate with microtubules during the entire maturation process, including spindle assembly, chromosome segregation, and polar body extrusion. Using drug treatments to disrupt the microtubule and actin spindles, we identify an important role of actin for microtubule-based spindle organization in human oocytes.

## Results

### Actin spans the spindle volume and associates with γ-tubulin.

We analyzed meiosis in human oocytes focusing our investigations on spindle assembly during the major stages of maturation. We set-up an experimental strategy based on multi-color 3D-fluorescence microscopy that allowed us to monitor the spatiotemporal organization of actin and microtubules simultaneously, starting from NEBD till the stage of metaphase II (MII) arrest. Using fluorescent phalloidin, SiR-actin, and actin-specific antibodies, our image analysis revealed that the spindle actin architecture in human MII oocytes is very similar to that of mouse oocytes[18]. Actin formed well-organized, parallel-arranged filamentous arrays that pervaded the entire spindle volume (Fig. 1a, b and Supplementary Movie 1). We observed single filaments and entire actin bundles that frequently aligned with microtubules and crossed the metaphase plate in the vicinity of chromosomes, occasionally touching them. At the spindle poles, actin filaments from the cytoplasm and the spindle interior congressed and formed a dense meshwork. We found a similar 3D organization of actin inside the spindle of developmentally disturbed oocytes that failed to extrude redundant chromosomes in meiosis I, and still contained the two sets of nuclear genetic material in meiosis II (Supplementary Fig. 1a). Moreover, we observed filamentous

actin structures penetrating the bipolar-shaped spindle-like microtubule assembly that surrounded sperm DNA (Supplementary Fig. 1b). Such structures are not typical but occasionally seen in intracytoplasmic sperm injection (ICSI)-treated oocytes. Thus, the co-assembly of the two cytoskeletal systems appears not to be restricted to the maternal microtubule spindle, but could also play a role during fertilization.

In oocytes of many species, the formation of a bipolar spindle proceeds through the self-organization of acentriolar microtubule organizing centers (MTOCs), which replace the function of centrosomes[19]. Human oocytes are in this respect different since they lack typical MTOCs[17]; however, they are still capable to configure a bipolar spindle and gather microtubules at the poles. To address whether actin might be involved in spindle assembly, we first tested in metaphase II oocytes for a co-localization with the γ-tubulin rich minus ends of microtubules, the origin of nucleation and growth initiation. We observed filamentous actin clusters overlapping at the spindle poles with distinct γ-tubulin foci (Fig. 1a, c and Supplementary Fig. 2b) at which single-microtubule fibers were attached (Supplementary Fig. 2a and Supplementary Movie 2). This is different from the localization typically observed in cumulus cells, where γ-tubulin shows a distinct centrosomal association (Supplementary Fig. 2c). Notably, multiple γ-tubulin foci were already observed in prometaphase II (Supplementary Fig. 2d). Given that the meiotic spindle in human oocytes progresses through highly instable multipolar and apolar stages till acquiring the typical bipolar configuration at metaphase II[17], the constant association of actin with γ-tubulin at the spindle poles suggests that actin may use γ-tubulin centers to establish the communication with microtubules and thus participate in the assembly of a functional bipolar spindle.

### Actin co-organizes with spindle microtubules during meiosis.

The close association of actin and microtubules at the spindle implies a functional inter-dependence between the two systems for proper spindle function, which we sought to investigate further. So far, actin and microtubules have only been visualized separately at the spindle[14,17], thus it remains unknown, how the two cytoskeletal structures spatiotemporally distribute and possibly interact with each other during meiosis. We addressed this aspect by visualizing the actin and microtubule cytoskeleton simultaneously in oocytes that were immature at the time of surgical retrieval following ovarian stimulation. The maturation of human oocytes from prophase I till the stage of MII arrest takes ~25 h[17]. We therefore allowed oocytes to resume maturation independent of any pharmacological trigger in vitro yielding oocytes with different maturation stages at the time point of fixation. The status of each oocyte could be assigned on the basis of (i) the presence or absence of a polar body to distinguish between early and later stages of meiosis and (ii) spindle architecture and chromosome configuration, both adopting characteristic morphologies in prophase, metaphase, and anaphase stages (Fig. 2a, b). We observed the first appearance of an organized actin assembly in the interior of the spindle in metaphase I. Actin displayed a barrel-shaped morphology similar to that of microtubules. Interestingly, in anaphase I, the stage of homologous chromosome separation, prominent filamentous actin structures filled the space between the separated chromosome sets and crossed the midzone area (Fig. 2b), however, without colocalizing with microtubules (Supplementary Fig. 3a and Supplementary Movie 3). During telophase I, spindle actin filaments penetrated the cortical membrane and protruded into the polar body separating the two chromatin masses in association with microtubules (Supplementary Movie 4). Actin and microtubules displayed a clear colocalization inside the polar body, suggesting

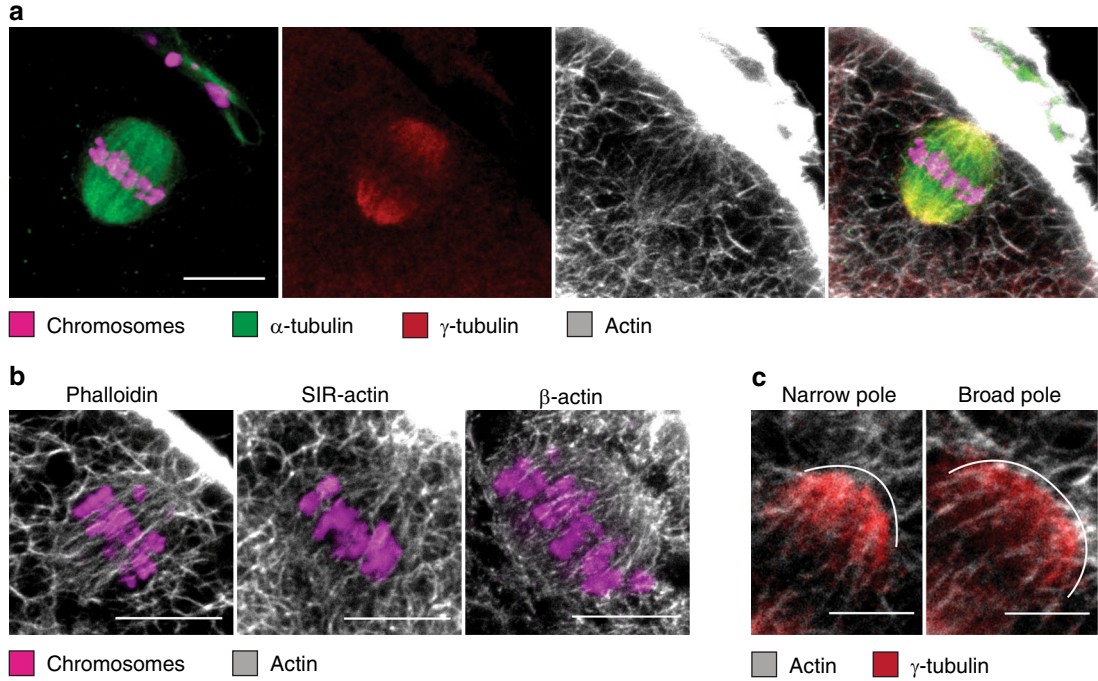

**Fig. 1** Filamentous actin spans the entire microtubule spindle volume in human metaphase II oocytes. **a** Representative z-projection of a metaphase II oocyte. Microtubules (α-tubulin), γ-tubulin, and actin (phalloidin) merge at the spindle poles; z = 6 sections. Representative of n = 5 oocytes. Scale bar, 10 μm. **b** Immunofluorescence projections of MII human oocytes stained for chromosomes (Hoechst) and filamentous actin visualized with fluorescent phalloidin, SiR-Actin, and anti-β-actin antibody; z = 6 sections. Representative of n = 14, 3, and 4 oocytes. Scale bars, 10 μm. **c** Close-up view of actin (phalloidin) and γ-tubulin distributions at narrow and broad spindle poles of metaphase II oocytes; z = 6 and 8 sections. Representative of n = 12 oocytes. Scale bars, 5 μm

that both proteins are responsible for the attachment of the decondensed chromatin to the polar body cortex (Supplementary Fig. 3b). This reveals that the interplay between actin and microtubules is not restricted to the spindle interior but also appears to be essential for keeping the genetic material separated during asymmetric division.

In prometaphase II, the stage of spindle decomposition and reassembly, only a few actin filaments penetrated the spindle and the poles were completely devoid of filamentous actin clusters (Fig. 2b). However, when microtubules acquired their bipolar organization in metaphase II, actin also reconstituted its typical barrel-like shape (Supplementary Movie 5). Altogether, these data reveal that actin is an integral component of the spindle that follows microtubule dynamics throughout the entire maturation process as schematically shown in Fig. 2a. Actin lost its ordered arrangement when microtubules decomposed and reacquired its well-defined organization during the reassembly of microtubules into a bipolar spindle. The pronounced presence of actin in the spindle midzone during anaphase I and telophase I, the stages of chromosome separation, suggests that actin might be involved in supporting microtubules during chromosome pushing to ensure that homologous chromosome pairs are kept apart until cytokinesis and polar body extrusion is successfully accomplished[20].

Experiments using antibodies directed against β-actin, γ-actin[21,22], and tropomyosin isoforms[23,24] revealed that the entire pool of actin, including cytoplasmic actin, spindle actin, and actin in the polar body was decorated with the tropomyosin isoforms Tpm3.1/3.2 (Supplementary Fig. 4). In metaphase II, where the spindle is stably attached to the cortex, spindle actin filaments facing towards the center of the oocyte contained predominantly the β-actin isoform, whereas the spindle actin filaments facing the inner cortical leaflet were mainly composed of γ-actin (Supplementary Fig. 5). We identified γ-actin as the predominant

isoform of cortical and subcortical actin structures (Supplementary Fig. 5a). Interestingly, both actin isoforms overlapped at the interface between cytoplasm and cortex forming a well-defined circular boundary (Supplementary Fig. 5b). In mouse oocyte, distinct pools of β- and γ-actin filaments determine polarity during asymmetric division[25]. These distinct localization patterns of the two actin isoforms might be required to coordinate the migration process by defining leading and trailing ends enabling a perpendicular orientation relative to the cortex during anchorage[26].

**Actin and microtubules cooperate at the meiotic spindle**. If actin and microtubules indeed do interact at the spindle, their dynamics should depend on each other. To test this assumption, we performed experiments involving acute drug addition to disturb one of both cytoskeletal structures and study the consequences for the other. Knowing that the impact of drugs on the cytoskeleton depends on incubation time and concentration[27–29], we tested different conditions until we were able to reproducibly affect the cytoskeleton without destroying the oocyte. Inhibition of the Eg5-mediated coherence between antiparallel microtubules through the addition of monastrol substantially altered spindle shape. This is highlighted in the loss of bipolarity and abnormal microtubule arrangements (Supplementary Movie 6). Interestingly, actin filaments mirrored this behavior as they still associated with the microtubules (Fig. 3a). Accordingly, stabilization of the bipolar spindle by taxol led to a drastic increase in spindle size, as reflected in elongated microtubules and actin fibers that emanated from pronounced actin clusters at the spindle poles and projected towards the spindle center (Fig. 3b, c). Altogether, these data demonstrate that the spatiotemporal organization of actin during oocyte maturation strictly follows microtubule dynamics.

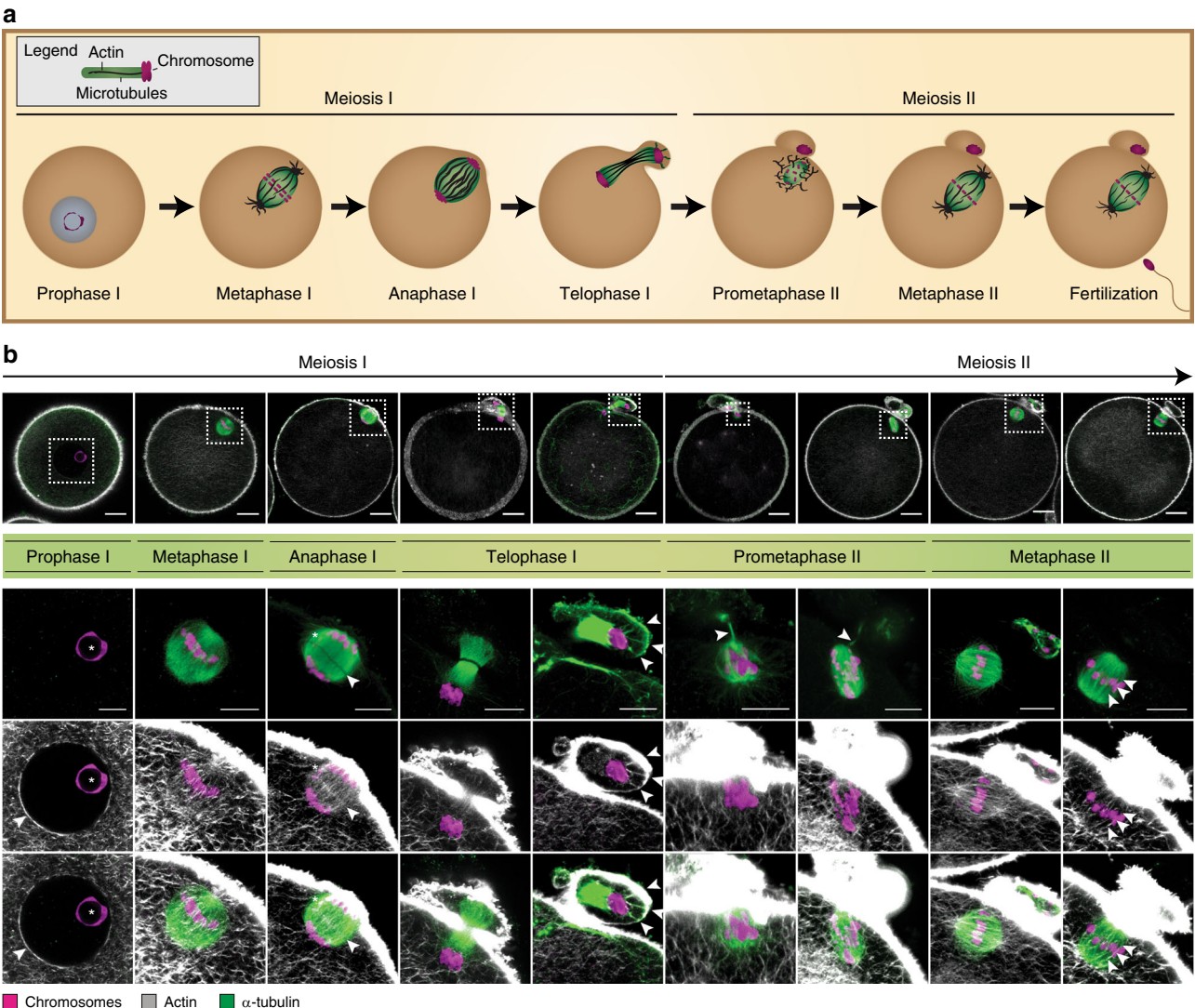

**Fig. 2** Actin and microtubule dynamics are tightly coupled during oocyte meiosis. **a** Schematic illustration of the characteristic stages of oocyte meiosis showing chromosome, microtubule, and actin organization at the spindle. **b** Confocal z-projections of oocytes fixed at different meiotic stages and stained for chromatin (Hoechst), actin (phalloidin), and microtubules (α-tubulin). z = number of projected sections; Images are representative of n = number of oocytes per stage. Prophase I: arrowhead points at the membrane of the germinal vesicle and asterisk marks chromatin clustering around the nucleolus; z = 3 sections, n = 10 oocytes. Metaphase I: z = 3 sections, n = 9 oocytes. Anaphase I: arrowhead highlights the spindle midzone, asterisk marks a lagging chromosome; z = 11 sections, n = 1 oocyte. Telophase I: arrowheads highlight regions of chromatin anchorage; z = 3 and 6 sections, n = 6 oocytes. Prometaphase II: arrowheads mark microtubules projecting towards the polar body; z = 8 and 11 sections, n = 7 oocytes. Metaphase II: arrowheads highlight actin fibers overlapping with microtubules; z = 6 and 8 sections, n = 15 oocytes. Scale bars are 20 μm (overview) and 10 μm (inset)

To validate this observation, we depolymerized microtubules with nocodazole, which resulted in multipolar spindles (Fig. 3d, f). A complete depletion of microtubules was not achieved, even when higher concentrations of the drug were used and incubation times increased. This was surprising, because in mouse oocytes nocodazole is not known to induce spindle multipolarity[14,30]. Longer incubations at higher doses increasingly reduced microtubule mass, affected chromosome alignment, and spindle shape (Supplementary Fig. 6a). Moreover, we detected distinct accumulations of γ-tubulin at each pole of the multipolar spindle (Fig. 3e, f). Notably, the chromosomes maintained their connection to microtubules and accumulated in small groups between the poles (Supplementary Movie 7). This phenotype was consistently observed for all nocodazole-treated oocytes, whereas it was hardly seen in untreated oocytes (Fig. 3d).

Importantly, nocodazole-induced microtubule destabilization, caused a gradual depletion of intra-spindle actin, and led to the loss of its typical bipolar organization indicating the close interplay between the two cytoskeletal systems (Fig. 3f and Supplementary Fig. 6b). Additionally, we treated oocytes with latrunculin-B prior to nocodazole addition to clarify if the disturbance of both cytoskeletal structures would have additional consequences on spindle architecture and chromosome clustering, which was not the case (Supplementary Fig. 6c). Altogether, these data indicate that microtubules might be responsible for dictating actin dynamics at the spindle.

To obtain a more detailed picture of the crosstalk between actin and microtubules, we exposed oocytes to either monastrol or nocodazole and subsequently incubated them in taxol. The subsequent use of destabilizing and stabilizing agents was a helpful tool to better visualize the strongly compromised cytoskeletal structures at the spindle poles after microtubule impairment and to validate our previous finding that the interplay between the two cytoskeletal systems is sensitive to

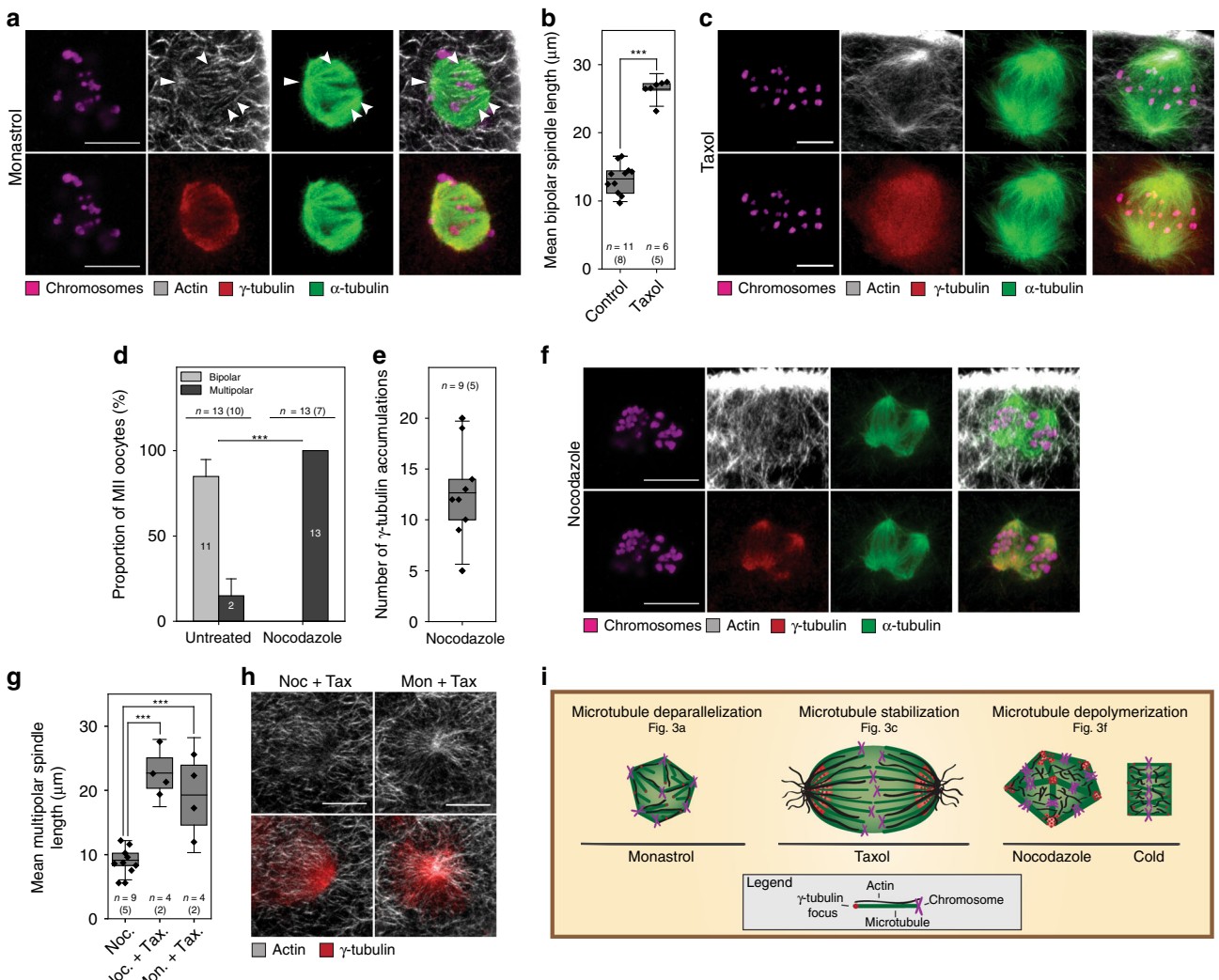

**Fig. 3** Disruption of microtubules promotes multipolar spindle formation and causes spindle actin reorganization. **a** Representative *z*-projections of monastrol-treated MII oocytes showing actin (phalloidin) and microtubules (α-tubulin) as indicated by arrowheads (upper panel); γ-tubulin and microtubule colocalization (lower panel); $z = 3$ sections. Representative of $n = 4$ oocytes. **b** Mean length of bipolar spindles in untreated and taxol-treated MII oocytes. **c** Localization of chromosomes (Hoechst), actin (phalloidin), α-tubulin (upper panel), and γ-tubulin (lower panel) in taxol-treated MII oocytes; $z = 27$ sections. Representative of $n = 6$ oocytes. **d** Frequency of bipolar and multipolar spindles in untreated and nocodazole-treated MII oocytes. **e** Number of distinct γ-tubulin accumulations in nocodazole-treated MII oocytes. **f** Immunofluorescence images showing chromosomes (Hoechst), actin (phalloidin), and microtubules (α-tubulin) at the multipolar spindles of oocytes treated with nocodazole (upper panel); γ-tubulin at multipoles (lower panel); $z = 10$ sections. Representative of $n = 5$ (upper panel) and $n = 4$ (lower panel) oocytes. **g** Mean distance between spindle multipoles in oocytes treated with nocodazole alone and oocytes consecutively treated with nocodazole and taxol or monastrol and taxol. **h** Immunofluorescence images showing actin (phalloidin) and γ-tubulin at the poles of multipolar spindles in MII oocytes consecutively treated with nocodazole and taxol or monastrol and taxol; $z = 5$ and 8 sections. Representative of $n = 4$ and 4 oocytes. **i** Schematic illustration of chromosomes, microtubules, and actin at the spindle of MII oocytes upon treatment with different cytoskeletal drugs or exposure to cold. Scale bars, 10 μm. $n =$ total number of oocytes. The number of donors is specified within parentheses. Center lines represent mean values; boxes illustrate 25th and 75th percentiles. Error bars represent standard deviation (**b**, **e**, **g**) and standard error of the mean (**d**). Significance was tested using Fisher's exact test (**d**) and the two-tailed two sample *t*-test (**b**, **g**). Asterisks denote *p*-values: *** < 0.001. Source data are provided as a Source Data file

the polymerization state of microtubules. Both procedures yielded large multipolar spindles (Fig. 3g and Supplementary Movies 8 and 9); however, a communication between actin and microtubules was only observed in monastrol-treated oocytes, where actin maintained its clear association with the spindle poles. This was reflected in the pronounced accumulation of filamentous actin structures radially spreading from the poles into the spindle interior (Fig. 3h and Supplementary Movie 10).

This artificially induced phenotype suggested that actin dynamics correlate with the dynamic instability of microtubules, which we intended to characterize further. We therefore exposed oocytes to cold, thereby inducing spindle shrinkage without affecting spindle bipolarity[14]. The shortening of the microtubules with time induced a broadening of the spindle poles, led to a decrease in spindle actin density, and complete loss of the ordered actin organization (Supplementary Fig. 7). Thus, both, cold and nocodazole treatments affected actin organization to a similar extent, highlighting the importance of an intact microtubule structure for a functional spindle actin organization. Altogether, with the results from the single and dual drug treatments, the data reveal a dominant role of microtubules for the communication of the two systems (Fig. 3i) and provide a possible explanation of how actin may be kept at the

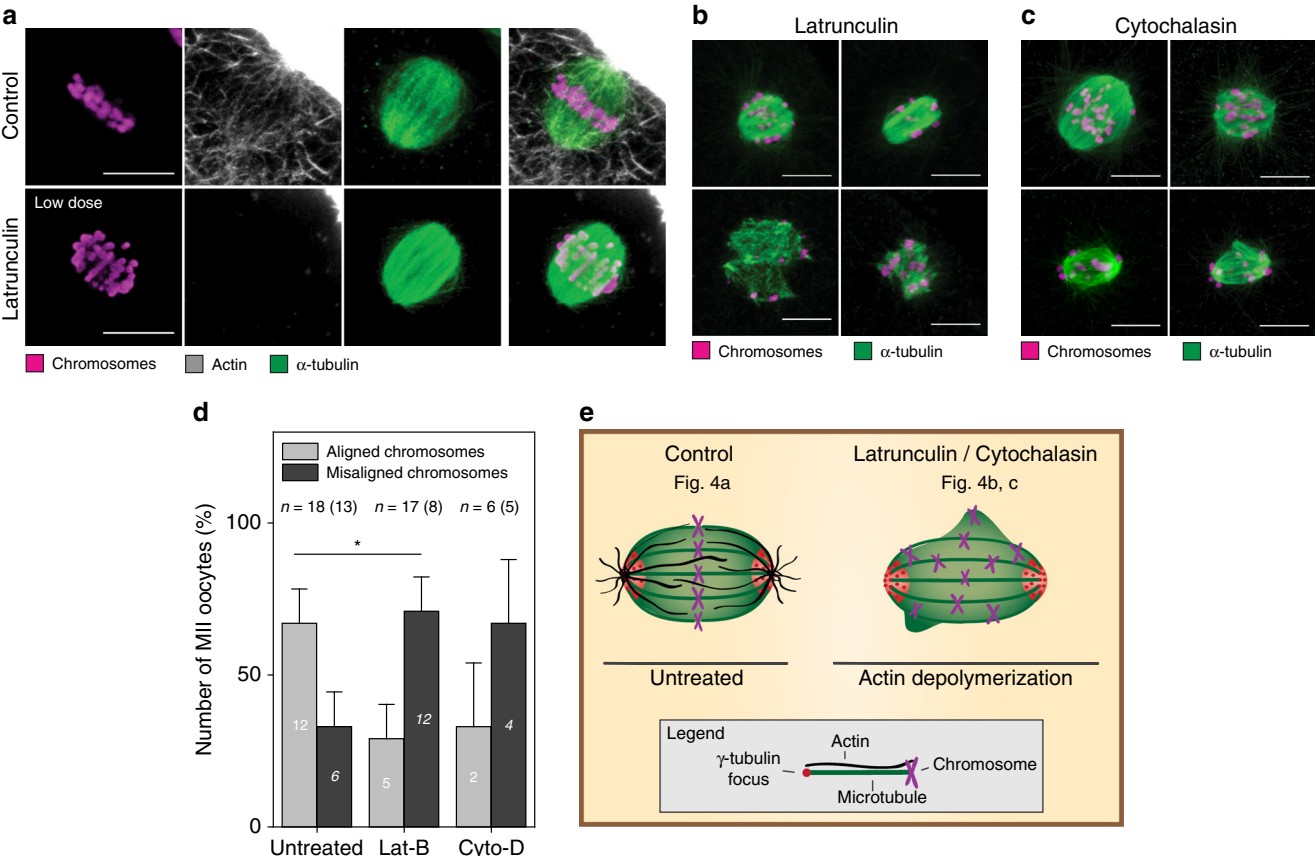

**Fig. 4** Disruption of filamentous actin affects chromosome alignment and perturbs bipolar spindle architecture. **a** Representative z-projections of untreated human MII oocytes (upper panel, same oocyte as shown in Fig. 1a) and oocytes treated with 2.5 μM latrunculin-B (lower panel) showing chromosomes (Hoechst), actin (phalloidin), and microtubules (α-tubulin); z = 16 and 11 sections. Representative of n = 15 and 3 oocytes. **b** Examples of MII oocytes treated with 1 μM (upper panel) or 5 μM (lower panel) latrunculin-B showing chromosomes (Hoechst), actin (phalloidin), and microtubules (α-tubulin); z = 4 sections. Representative of n = 4 (upper panel) and n = 8 (lower panel) oocytes. **c** Examples of MII oocytes treated with 5 μg/ml (upper panel) or 10 μg/ml (lower panel) cytochalasin-D showing chromosomes (Hoechst), actin (phalloidin), and microtubules (α-tubulin); z = 4 sections. Representative of n = 4 (upper panel) and n = 2 (lower panel) oocytes. **d** Comparison of oocytes with aligned vs unaligned chromosomes in the absence or presence of actin depolymerizing agents (1, 2.5, 5 μM latrunculin-B incubations for 5 to 60 min prior to fixation; 5 μg/ml and 10 μg/ml cytochalasin-D for 25 and 60 min prior to fixation). **e** Schematic illustration of chromosomes, microtubules, and actin at the spindle of MII oocytes upon treatment with actin depolymerizing agents. Scale bars, 10 μm. n = total number of oocytes. The number of donors is specified within parentheses. Error bars represent standard error of the mean. Significance was tested using Fisher's exact test. Asterisk denotes p-value: * < 0.05. Source data are provided as a Source Data file

spindle for safeguarding the chromosome segregation process during all subsequent stages of meiosis.

**Actin assists microtubules in spindle assembly.** Next, we addressed whether actin disruption may also have an influence on microtubules. The addition of latrunculin-B abolished actin at the spindle with time but the spindle architecture remained unaffected only at low concentrations (Fig. 4a). Increased concentrations (5 μM) induced severe spindle disturbances within minutes. Some oocytes displayed a multipolar-like spindle geometry, in which microtubules were completely misarranged (Fig. 4b, e). To reassure the effects of actin depletion on spindle morphology and chromosome alignment, we chose an additional approach based on cytochalasin-D to disrupt actin through a different mechanism[31]. We observed that some chromosomes failed to maintain their metaphase plate alignment or were scattered across the spindle volume (Fig. 4c, e). This is consistent with previous results, where actin depletion had drastic effects on chromosome alignment during metaphase II in mouse oocytes[14]. The disarrangement of chromosomes could result from weakened chromosome attachments or spindle instabilities, including loss

of spindle bipolarity[14,17]. Careful inspection of the untreated and latrunculin-treated oocytes allowed us to estimate how severely actin depletion affected chromosome alignment. We quantified the portion of oocytes containing misaligned chromosomes, i.e., more than two chromosomes localizing outside the metaphase plate or scattering around the poles. In the untreated group, the majority of oocytes (12/18) displayed well aligned chromosomes and normal spindle architecture. Notably, the proportion of intact chromosome arrangements was reduced, when oocytes were treated with latrunculin (5/17) or exposed to cytochalasin (2/6) (Fig. 4d). These data agree with the previous finding that actin is essential for chromosome alignment in mouse oocytes providing further support for a conserved role of actin at the meiotic spindle throughout mammalian species[32]. It has been proposed that in human oocytes, contrary to mice, spindle assembly is realized in an extremely slow process involving subsequent stages of instable multipolar spindle intermediates, which are thought to increase the incidence of chromosome segregation errors[17]. The fragmentation of the poles into multipoles upon the addition of nocodazole observed herein supports the previous findings of an inherently instable human oocyte spindle and demonstrates

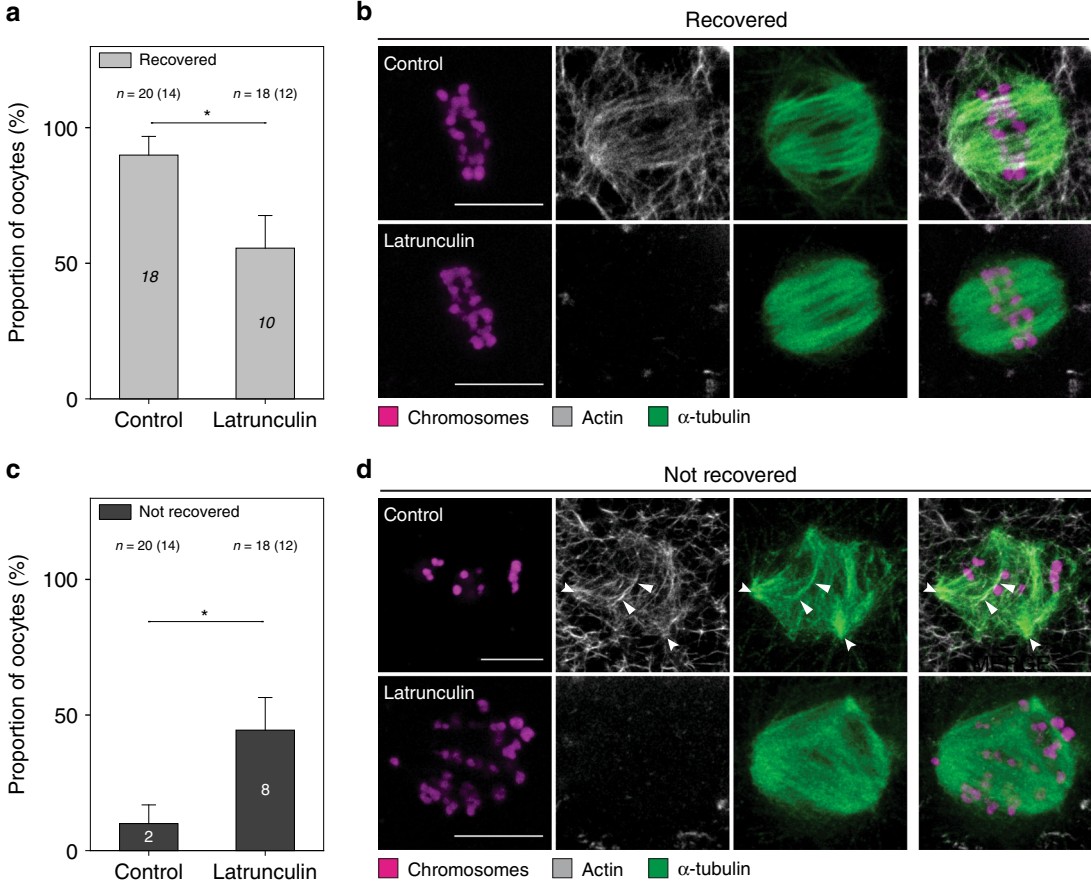

**Fig. 5** Actin supports microtubules in the assembly of a bipolar spindle. **a** Proportion of oocytes that restored bipolar spindle morphology (recovered) in the absence or presence of 1 µM latrunculin-B after nocodazole treatment. **b** Representative z-projections of recovered MII oocytes after nocodazole treatment and 60 to 90 min regeneration time in the absence or presence of 1 µM latrunculin-B showing chromosomes (Hoechst), α-tubulin, and actin (phalloidin); z = 4 sections. Representative of n = 15 (upper panel) and n = 10 (lower panel) oocytes. **c** Proportion of oocytes that retained spindle multipolarity (not recovered) in the absence or presence of 1 µM latrunculin-B after nocodazole treatment. **d** Representative z-projections of not recovered MII oocytes after nocodazole treatment and 60 to 90 min regeneration time in the absence or presence of 1 µM latrunculin-B showing chromosomes (Hoechst), α-tubulin, and actin (phalloidin). Arrowheads highlight actin-microtubule overlap; z = 4 and 5 sections. Representative of n = 2 (upper panel) and n = 8 (lower panel) oocytes. Scale bars, 10 µm. n = total number of oocytes. The number of donors is specified within parentheses. Error bars represent standard error of the mean. Significance was tested using Fisher's exact test. Asterisks denote p-values: * < 0.05. Source data are provided as a Source Data file

further that spindle instability is preserved in the mature, MII arrested stage, potentially influencing chromatid partitioning during the second meiotic division.

To support the assumption of an active involvement of actin in spindle assembly, we performed additional drug addition experiments that we designed to affect the dynamics of the two cytoskeletal systems transiently and subsequently. This indirect approach was necessary, since, e.g., a specific disruption of actin in the spindle interior for investigating functional consequences is difficult to realize, particularly in light of restricted methods for selective manipulation of intracellular structures in human-derived material and ethical concerns. We took advantage of the multipolarity-inducing effect of nocodazole (Fig. 3f) and allowed MII oocytes to reform a bipolar spindle from the multipolar state while keeping actin unaffected or depleted (Fig. 5 and Supplementary Fig. 8). Oocytes with an intact actin cytoskeleton recovered spindle bipolarity more frequently than oocytes with disrupted actin (Fig. 5a, b). In the few cases, in which recovery was not achieved within the defined time window, we observed actin filaments closely associating with microtubules in the near vicinity of chromosomes (Fig. 5d). Even under these circumstances, actin and microtubules appear to share similar dynamics, which is indicative of a decisive role of actin for spindle

formation. Accordingly, actin-depleted oocytes failed more frequently to assemble a bipolar spindle at the time of inspection than untreated oocytes (Fig. 5c).

## Discussion

The identification of actin inside the spindle and its active contribution in chromosome segregation previously reported for the mouse system[33] have challenged our understanding of the mechanisms underlying oocyte meiosis. Here, we established that actin is an essential component of the human oocyte spindle throughout the entire oocyte maturation process, where it accompanies microtubule dynamics during spindle formation and asymmetric division in meiosis I, and also during the subsequent stages of spindle decomposition and spindle reconstruction in meiosis II. The recruitment of actin to the spindle and its co-organization with γ-tubulin clusters appear to be strictly associated with the spatiotemporal rearrangement and reassembly microtubules undergo during oocyte maturation (Figs. 2 and 3). Our data indicate that actin and microtubules cooperate in assembling a functional acentrosomal spindle, which progresses through apolar and multipolar intermediates[17]. In this respect it appears that microtubules and their changing dynamics

are an important factor dictating the association of actin with the spindle, where it is held in place to safeguard the most critical steps of human oocyte meiosis, i.e., the assembly of a functional bipolar spindle and accurate chromosome segregation.

The data are consistent with the emerging picture that actin and microtubules are mechanically coupled supporting each other's function[34]. Additionally, the colocalization during polar body extrusion hints at a joint role of actin and microtubules in keeping chromatin masses apart, until the division process is successfully accomplished. In humans, where oocyte aneuploidy is exceptionally frequent and accounts for the majority of miscarriages and congenital diseases[35,36], the herein observed interplay may reflect a mechanism used to compensate for the lack of centriolar microtubule organizing centers important to facilitate spindle assembly and chromosome sorting during the different stages of oocyte meiosis.

Age-related changes in microtubule acetylation have been reported to directly affect spindle function[37,38]. In light of the critical role of posttranslational modifications on microtubule stability, similar mechanisms altering actin dynamics, including the regulation by tropomyosin copolymers may also affect spindle stability with implications for fertility and embryo development. In summary, our findings open perspectives for future investigations addressing the regulatory mechanisms underlying the crosstalk between actin and microtubules during meiosis and subsequent stages of embryo development, which involves the formation of dual-spindles in the transition from meiosis to mitosis[39].

## Methods

**Ethics statement**. The use of human oocytes in this study has been approved by the ethics committee of Hannover Medical School (MHH) under the registration number 6251. Donating women gave informed consent for the use of their oocytes in this study after oral and written elucidation on all aspects of the scientific project and their legal rights, including the right to withdraw the consent at any time without negative consequences. The women were not offered any reward for donation. The study complies with the legal regulations of the German Act for Protection of Embryos (Gesetz zum Schutz von Embryonen).

**Collection and preparation of human oocytes**. A total of 604 human oocytes were collected from 184 women receiving assisted reproductive treatment (ART) between October 2016 and May 2019. All oocytes donated for this study were retrieved to employ ICSI. Couples were assigned to ICSI treatment because of male factor infertility. Meiotic stages were investigated with oocytes that failed to resume meiosis following ovarian stimulation and were in an immature germinal vesicle stage at the time of surgical retrieval, thus unsuitable for sperm injection. These oocytes received no ICSI treatment and subsequently underwent spontaneous maturation in vitro for 20 to 26 h independent of a pharmacological trigger till they were fixed. All other experiments were performed with MII oocytes that received ICSI treatment but had not formed pronuclei at the time of donation (24 h after sperm injection). Only oocytes with no signs of perturbation, like degeneration or other morphological features indicative of poor quality by light microscopic inspection were included in the study independent of the condition of the donors in terms of health, age, lifestyle, and hormonal stimulation, all of which contribute to oocyte quality. Therefore female factors affecting oocyte quality cannot be excluded. Experiments were started immediately after donation, about 57.5 to 61.5 h after hCG injection used to trigger maturation in vivo. Fixation and preparation was performed as follows: the cumulus oophorus cells were detached from the oocytes after short incubation (max. 90 s) in flushing medium [Origio 10845060A] containing 80 U/ml of hyaluronidase [Serva 25118]. Following ICSI, MI, and MII oocytes were cultured in G1-Plus medium [Vitrolife 10128] in a CO$_2$-incubator [Labotec C200] with 6.0% CO$_2$, 92% RH, and 9% O$_2$ at 37 °C. Oocytes in the germinal vesicle stage were also kept in G1-Plus medium [Vitrolife 10128]. The zona pellucida (ZP) was removed by placing oocytes in a 60 µl drop of acidic Tyrode's solution [Sigma-Aldrich T1788]. Dissolving of the zona pellucida (30 s to 1 min) was monitored under the stereo microscope [Wild Heerbrugg Type 346910 or Olympus SZ-STS] and oocytes were then shortly washed in a 60 µl drop of HEPES-HTF solution [Irvine Scientific 90126] containing 0.3% bovine serum albumin (BSA). To avoid evaporation all drops were covered with light mineral oil [MERCK Millipore ES-005-C] and kept at constant temperature (37 °C). Oocytes were fixed and permeabilized in one-step using a 100 mM PIPES [AppliChem A1079] solution (pH 7.0) containing 5 mM magnesium chloride hexahydrate [MERCK 105832], 2 mM EGTA [Roth 3054.3], 2% formaldehyde [Sigma-Aldrich F1635], 0.5% Triton X-100 [MERCK 108603], and 10 nM paclitaxel [Sigma–Aldrich T7191] for 30 min at 37 °C. Then, oocytes were washed two to

three times in 40 µl drops of phosphate buffered saline (PBS) [biowest L0615-500] blocking solution containing 2% BSA [Sigma A4503], 2% fetal bovine serum [Sigma 7524], 100 mM Glycin [Roth 3790], and 0.01% Triton X-100 [MERCK 108603] for 5 min followed by an additional 1 h incubation in blocking solution. All washing and blocking steps were performed at room temperature. Subsequently, oocytes were either stained or kept in blocking solution at 4 °C for later staining. Oocytes stained for β- and γ-actin were fixed, washed, and blocked in the absence of Triton X-100 and membrane permeabilization was performed in 100% ice-cold methanol (−20 °C) for 5 min.

**Confocal microscopy and 3D imaging**. Depending on the experiment, oocytes were incubated with different primary and secondary antibodies either at room temperature for 1 h or at 4 °C overnight. Incubation with secondary antibodies was performed at room temperature for 1 h. Microtubules were stained using primary anti-alpha-tubulin monoclonal antibody [1:50 or 1:100, Molecular Probes A11126] with either secondary goat anti-mouse Alexa Fluor 488 [1:200, Molecular Probes A11029], goat anti-mouse Alexa Fluor 647 [1:200, Molecular Probes A21235] or goat anti-mouse IgG1 Alexa Fluor 488 [1:200, Jackson Immunoresearch 115-545-205] antibodies. Gamma-tubulin was stained using primary anti-gamma-tubulin antibody [1:100, Santa Cruz Biotechnology sc17788] and secondary goat anti-mouse IgG2b Cy5 antibody [1:100, Jackson Immunoresearch 115-175-207]. Tropomyosin 3.1/3.2 was stained using primary polyclonal anti-γ9d antibody [1:200, Merck Millipore AB5447] and secondary donkey anti-sheep Cy3 antibody [1:200, Merck Millipore AP184C]. Actin was stained with Alexa Fluor Phalloidin 488 [Molecular Probes A12379], Alexa Fluor Phalloidin 568 [Molecular Probes A12380], or Alexa Fluor Phalloidin 647 [Molecular Probes A22287] at concentrations of 1 U/l. For the visualization of actin with SiR-Actin [Cytoskeleton CY-SC001], a concentration of 100 nM was used. Spindle actin was also stained using primary anti-beta-actin antibodies [1:50, Nordic MUbio MUB0110P] and secondary goat anti-mouse IgG1 Alexa Fluor 488 antibody [1:200, Jackson Immunoresearch 115-545-205] or primary anti-gamma-actin antibody [1:100, Nordic MUbio MUB0111P] and secondary goat anti-mouse IgG2b Cy5 antibody [1:100, Jackson Immunoresearch 115-175-207]. All antibodies were diluted in PBS containing 5% BSA [Sigma A4503]. DNA was stained with 5 µg/ml Hoechst 33342 [Thermo Fisher 62249]. For microscopy, oocytes were mounted in a chamber composed of two coverslips separated by a double-sided adhesive tape [Tesa 05338]. This prevented squeezing of the oocyte and allowed imaging from both sides of the chamber increasing the probability to find the spindle area close to the objective avoiding problems in image quality due to limitation in penetration depth. Oocytes were imaged using a Leica TCS SP8 microscope equipped with a 63x PL APO CS 1.4 NA oil immersion objective lens [Leica, Wetzlar]. Confocal z-stacks were obtained from single images taken every 0.5 µm.

**Experiments with drugs and cold treatment**. For experiments targeting actin and microtubule dynamics, oocytes were treated with different drugs diluted in HEPES-HTF [Irvine Scientific 90126] containing 0.3% BSA [Sigma A4503]. To disturb cohesion among antiparallel microtubules, oocytes were treated with 100 µM monastrol [M8515 Sigma-Aldrich] for 1 h. To induce microtubule depolymerization, nocodazole [M1404 Sigma-Aldrich] was used at a concentration of 25 µM for 25 min. To stabilize microtubules, oocytes were incubated with 50 µM taxol [Sigma–Aldrich T7191] for 1 h. Actin depolymerization was induced using 1, 2.5, and 5 µM latrunculin-B [L5288 Sigma-Aldrich] or 5 µg/ml and 10 µg/ml cytochalasin-D [C8273 Sigma-Aldrich] for 5 to 60 min depending on the experiments. An incubation time of 25 min with 2.5 µM was enough to depolymerize spindle actin and most of the cytoplasmic actin leaving cortical actin mostly unaffected. For cold-induced microtubule destabilization oocytes were placed in a petri dish covered with 7 ml of warm (37 °C) light mineral oil [Zenith Biotech ZPOL-500] and left to cool down at 4 °C for 6 to 15 min. All drug and cold treatments were performed prior to zona pellucida removal.

**Image and data analysis**. Images of filamentous actin at the spindle are average intensity projections. All other channels were obtained following maximum intensity projection of the stacks. Image processing and analysis were performed using ImageJ software and Imaris [Bitplane]. Colocalization was evaluated using Imaris section view and the Coloc2 algorithm in ImageJ. Meiotic stages have been assigned according to (i) the absence or presence of a polar body, (ii) configuration of the chromosomes, (iii) spindle morphology and spindle position. Spindle lengths were measured with Imaris using the measurement point tool. The number of γ-tubulin accumulations was quantified using the spot detection tool in Imaris and approved manually based on the intensity, size and connection to spindle microtubules. The multipolar spindle length was calculated as mean distance between γ-tubulin accumulations using Imaris and the pdist command in MatLab [MathWorks].

**Statistics**. Quantification, statistical evaluation, and preparation of graphs were performed with Origin software [OriginLab]. Mean values, standard deviation, and statistical significance were evaluated with the two sample t-test (two-sided). In box

plots, center lines represent mean values, error bars show standard deviations and boxes illustrate 25th and 75th percentiles. For experiments with binary outcome, Fisher's exact test was used to test for significant differences between groups. Here, error bars represent standard error of the mean. Asterisks are used to denote $p$-values: $* < 0.05$, $** < 0.01$ and $*** < 0.001$. $n$ declares the number of oocytes used, the number of donors is stated within parentheses.

**Reporting summary**. Further information on research design is available in the Nature Research Reporting Summary linked to this article.

## Data availability

The authors declare that the data supporting the findings of this study are available within the paper and its supplementary information files.

The source data underlying Figs. 3b, d–e, g, 4d, 5a, c and Supplementary Fig. 7b are provided as a Source Data file.

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

## Acknowledgements

We are grateful to the clinicians and the embryology team at the Deutsche Klinik Bad Münder, the clinical directors Arvind Chandra and Elmar Breitbach, and the head of the IVF laboratory Uwe Pohler. We thank Constantin von Kaisenberg for establishing the contact to the Deutsche Klinik Bad Münder and Sharissa Latham for help and discussions. We thank all women who donated their oocytes. J.R. was supported by the Hannover Biomedical Research School (HBRS) as part of the StrucMed Program.

## Author contributions

G.T. initiated and advised the study; J.R. performed the experiments; J.R. and G.T. designed experiments and analyzed data; G.T. supervised and advised on the design of the experiments; J.R. and G.T. wrote the manuscript.

## Competing interests

The authors declare no competing interests.

## Additional information

**Peer Review Information** *Nature Communications* thanks the anonymous reviewers for their contribution to the peer review of this work. Peer reviewer reports are available.

