## [Peer Review File · Nature Communications]

Reviewers' comments:

Reviewer #1 (Remarks to the Author):

The manuscript provides a detailed description of the meiotic spindle and the surrounding actin filament network in human oocytes. The main novelty is that the authors show for the first time that an 'actin spindle' exists in human oocytes, very similar in morphology and very likely with similar function as recently described in mouse oocytes by the Schuh laboratory.

The work is very carefully executed, the immunofluorescent stainings are of high quality providing a very detailed description of steps of meiotic divisions in human oocytes. The functional experiments are somewhat limited, but this is inevitable due to the limited availability of human oocytes.

Overall, the manuscript is very well written and clear, and at the same time rather short and descriptive, offering little mechanistic insight. However, given the high quality of the data and the exceptional relevance of the findings to human reproduction, I would recommend the manuscript for publication in Nature Communications.

I only have minor comments:

1. For drug treatments it should be clearly stated that most of these treatments achieve only partial inhibition. For example, in nocodazole treated oocytes a microtubule spindle is still present, clearly evidencing that microtubules have not been completely depolymerized.
2. Even more important is that these results are interpreted cautiously considering these partial effects. In my opinion combinations of such partial effects are generally very difficult to interpret, and therefore I would even consider removing some of those experiments in which multiple drugs have been added to the oocytes.
3. Without careful controls and more detailed characterization I am not sure that the anti-actin and anti-tropomyosin antibody stainings are particularly informative, as these antibodies are known for their tendency to stain off targets. In my opinion phalloidin and Lifeact already provides sufficient evidence for the presence of the observed F-actin structures.

Reviewer #2 (Remarks to the Author):

Comments to the authors

The authors report that the existence of actin spindles during meiosis I and II in human oocytes. This first becomes evident in late meiosis I and the actin fibres appear between the two masses of chromatin during anaphase I. Microtubule spindle reassembly at meiosis II appears to be accompanied by reassembly of the actin spindle. Using drug treatments to disrupt the microtubule and actin spindles, the authors conclude that actin spindle dynamics follow those of the microtubule spindle.

The finding that actin infiltrates the spindle in human oocytes is consistent with a previous report, which included mouse and human oocytes (Mogessie and Schuh, 2017, Science). The current manuscript builds on this work by further characterising the behaviour of actin fibres during progression through meiosis in human oocytes.

Major comments

The study is based on oocytes that were either not suitable for use in ICSI treatment, or failed to undergo fertilisation. A major caveat, not mentioned by the authors, is that these oocytes may be inherently abnormal and may therefore not provide an accurate picture of how healthy human oocytes progress through the meiotic divisions. In addition, the authors make no effort to explain

the provenance of the oocytes used in each experiment. This makes it very difficult to judge the quality and reliability of the data. For example, it is not at all clear what criteria were used to assign meiotic stages to the fixed oocytes shown in Fig 1. Were the oocytes matured in vitro and fixed at specific timepoints? Or were the stages assigned based on the configuration of the chromosomes and spindle. If the latter, how can one distinguish between a prometaphase oocyte and one that simply failed to assemble a spindle, which would not be unexpected for this source of oocytes. It would therefore be important to clarify the status of oocytes used for each experiment, if possible providing the time from administration of the hCG injection used to trigger maturation in vivo.

A further consideration in relation to the supply of oocytes, is that the ethics committee approval was granted for the use of unfertilised eggs (Methods section), yet it is clear from the data that some eggs contained sperm chromatin. Strictly speaking, these eggs are not unfertilised. Indeed, in some jurisdictions they would be legally considered as embryos. The authors, should therefore provide further clarification on the ethics approval to use eggs that contain sperm chromatin, but (presumably) do not form pronuclei.

Minor comments

The manuscript suggests that mouse oocytes at the metaphase II stage do not have a multi-polar spindle, but the cited article shows that this is not the case; though the number of foci in the example images from the cited paper and this manuscript suggest human metaphase II oocytes have more foci. Due to the nature of fixed data, it is surprising that the authors find their data support a case for multipolar intermediates in human oocyte spindle assembly; especially as Figure 1 suggests otherwise.

It is surprising that culturing any cell containing a spindle for 25 minutes in 25uM nocodazole does not completely obliterate the microtubule spindle. Given the unexpected result, it would have been reassuring to see experiments in mouse oocytes using the same stock of nocodazole. Similarly, for the higher concentration of latrunculin B used to disrupt the actin cytoskeleton.

In addition, the authors should also consider the possibility of artefacts due to the addition of paclitaxel in the fixative. This would be expected to have a similar effect to taxol treatment (though the concentration is considerably lower).

The implication in the methods section is that MI oocytes were injected with sperm for ICSI treatment. This would not be considered normal practice. In addition, the authors comment on a second spindle forming around the paternal genome. Presumably, this observation was made in eggs that failed to form pronuclei 24 hours after sperm injection. If so, the authors should mention that it may not be reflective of normal progression thought fertilisation.

A final comment would be that it seems unusual that the authors used 9% O₂ to culture in media that is typically used at either atmospheric or 5% O₂.

The title of the manuscript seems overblown: the authors do not provide evidence to support the statement that actin drives meiosis in human oocytes.

Similarly, I am not convinced that the following sentence in the concluding paragraph is supported by the evidence presented in the manuscript - "the herein observed tight interplay between actin and spindle microtubules provides a mechanistic basis for how actin accompanies the process of spindle migration, anchorage, and chromosome segregation for successful oocyte maturation"

In summary, the authors deserve credit for studying such a large number of human oocytes, and for the very nice imaging. However, as discussed above, the value to the scientific community will be limited without further information on the oocytes used in each experiment.

Reviewer #3 (Remarks to the Author):

This manuscript describes the interplay between actin and microtubules in the meiotic spindle in human oocytes. First, the authors confirmed that actin forms a spindle-like structure tightly associated with spindle microtubules in human oocytes, as previously observed (Mogessie and

Schuh, Science 2017). They then further investigated the details of the actin spindle, providing some new observations of actin distributions during anaphase I and telophase I, and identifying tropomyosin and actin isoforms as components of the actin spindle. To address whether microtubules regulate the actin spindle, they treated human oocytes with drugs and conditions that affect microtubule dynamics. These experiments showed that the structural integrity of the actin spindle depends on microtubules, consistent with previous observations in mouse oocytes (Mogessie and Schuh, Science 2017). Finally, they addressed whether actin regulates the microtubule-based spindle, by perturbing actin dynamics during spindle reformation after nocodazole washout. The results appeared to suggest the importance of actin for microtubule-based spindle organization in human oocytes, which is again in agreement with the previous findings that actin contributes to spindle microtubule dynamics in mouse oocytes (Mogessie and Schuh, Science 2017).

Overall, the manuscript well describes the actin spindle and its functional relationships with microtubules in human oocytes. The observations presented in this manuscript are largely in line with what have been reported in mouse oocytes, therefore novelty is limited here. However considering that significant differences between mouse and human oocytes have been reported, datasets from studies using human oocytes are very important for the understanding of egg aneuploidy, the leading cause of pregnancy loss and several congenital disease. The image dataset and analysis of human oocytes presented here are of high quality, and therefore precious and likely attract readers from the fields of meiosis and reproductive biology and medicine. I would support publication of the manuscript in Nature Communications if the authors appropriately address my comment below.

Major comment:

- The number of oocytes tested in the experiment shown in Figure 3 is too small to make any clear conclusions. The authors should repeat the experiment to have a number of oocyte sufficient to test a significant difference between the two groups. This is critical because this experiment addresses the functional dependency of microtubules on actin in this manuscript. Conclusive data should be provided to support their claim of 'Actin-microtubule interplay'.
- To convince the contribution of actin to the microtubule spindle, it is recommended that authors add an experiment using another perturbation of actin dynamics. Currently the manuscript uses only Latrunculin, suggesting that spindle reformation is Latrunculin-sensitive. This may imply that actin contributes to spindle formation, but a concern here is a secondary effect of Latrunculin. If consistent results could be obtained by another perturbation (e.g. Cyto B or D, other inhibitors or RNAi?), their claim of 'Actin-microtubule interplay' would be further supported.

We thank the reviewers and the editor for their friendly evaluation of the manuscript, constructive comments and valuable suggestions to improve the quality of our work. We have addressed all points raised by the referees by including additional controls, validation experiments, and quantification data where appropriate. All major modifications made to the manuscript are highlighted in blue. We provide also a clean copy to the submission.

Reviewer #1 (Remarks to the Author):

The manuscript provides a detailed description of the meiotic spindle and the surrounding actin filament network in human oocytes. The main novelty is that the authors show for the first time that an ‘actin spindle’ exists in human oocytes, very similar in morphology and very likely with similar function as recently described in mouse oocytes by the Schuh laboratory.

The work is very carefully executed; the immunofluorescent stainings are of high quality providing a very detailed description of steps of meiotic divisions in human oocytes. The functional experiments are somewhat limited, but this is inevitable due to the limited availability of human oocytes.

Overall, the manuscript is very well written and clear, and at the same time rather short and descriptive, offering little mechanistic insight. However, given the high quality of the data and the exceptional relevance of the findings to human reproduction, I would recommend the manuscript for publication in Nature Communications.

I only have minor comments:

1. For drug treatments it should be clearly stated that most of these treatments achieve only partial inhibition. For example, in nocodazole treated oocytes a microtubule spindle is still present, clearly evidencing that microtubules have not been completely depolymerized.

We agree with the reviewer that our drug treatment experiments achieved only a partial inhibition, particularly in the case of nocodazole, where even longer incubation times (up to 60’) and higher concentrations (up to 100 μM , see **Supplementary Figure 6a** below) did not remove microtubules completely but led to a further reduction in spindle size, microtubule mass and disturbances in microtubule assembly. Importantly, in all cases nocodazole addition induced multipolarity in which spindle actin filaments increasingly lost their typical bipolar organization indicating the close interplay between the two cytoskeletal systems.

Supplementary Figure 6a | Effect of nocodazole on spindle microtubules in human oocytes. Immunofluorescence projections exemplifying the arrangement of chromosomes and microtubules in nocodazole-treated oocytes. $z = 11$ sections of $0.5 \mu\text{m}$. Scale bars, $10 \mu\text{m}$.

This data we included into the manuscript as follows:

...“A complete depletion of microtubules was not achieved, even when higher concentrations of the drug were used and incubation times increased. This was surprising, because in mouse oocytes nocodazole is not known to induce spindle multipolarity^{14, 30}. Interestingly, longer incubations at higher doses increasingly reduced microtubule mass, affected chromosome alignment, and spindle shape (Supplementary Fig. 6a).”...

Our control experiments in HeLa cells (**Figure A**) demonstrate the effectiveness of nocodazole to destabilize microtubules and reveal further that the potency of inhibition can vary between cell systems. HeLa cells receiving the same treatment lost all microtubule structures, even those of the mitotic spindle. Please find below some examples of nocodazole treated oocytes at different concentrations and incubation times.

Figure A | Effect of nocodazole on microtubules in HeLa cells. Representative immunofluorescence images of HeLa cells treated with 5, 10, 25 µM nocodazole or 0.15% DMSO for 25 minutes showing DNA, α-tubulin and γ-tubulin. Data are from three independent experiments. $z = 5$ sections of $0.3 \mu\text{m}$. Scale bars, $10 \mu\text{m}$.

2. Even more important is that these results are interpreted cautiously considering these partial effects. In my opinion combinations of such partial effects are generally very difficult to interpret, and therefore I would even consider removing some of those experiments in which multiple drugs have been added to the oocytes.

We agree with the reviewer that drug-induced effects are generally difficult to interpret, particularly if the targets are only partially compromised by the drug. Although single treatments did not abandon microtubules or actin completely, each drug had under the specified conditions (concentration, time of exposure) its characteristic and reproducible effect on one or even both cytoskeletal structures as manifested in changes in spindle polarity, spindle morphology, and chromosome alignment. These effects even persisted, when the cytoskeletons were increasingly deteriorated (**see also in manuscript Figs. 3, 4**), however, stronger impairments by higher latrunculin or cytochalasin concentrations and long-time exposures led to membrane deformation and increasingly affected overall oocyte shape (**Fig. B**). This situation does not allow conclusive interpretation of a functional interplay between the systems. We therefore used low doses of drugs in our experiments.

Regarding the question about the multi-drug experiments, we agree with the referee that interpretation of effects is not straightforward. We performed dual-drug treatments for different reasons:

- a) to clarify whether the simultaneous disturbance of the cytoskeletons (**Supplementary Figure 6c**) would induce additional changes in spindle architecture other than the multipolarity observed with nocodazole alone, which was not the case. Together with the single-drug treatments, the results reveal a dominant role of microtubules in the maintenance of a functional spindle;
- b) the subsequent treatment with taxol in the nocodazole and monastrol experiments was a helpful tool to improve imaging of the actin structures at the spindle poles. This further supported our previous finding (experiments without taxol) that the interplay between the two cytoskeletal systems is sensitive to the polymerization state of microtubules. This is reflected in the almost complete absence of actin from the poles in nocodazole/taxol experiments. The loss of microtubule coherence alone by Eg5-inhibition maintained the communication as shown by the strong and well-defined association of actin with the spindle poles.
- c) Finally, the dual-drug treatment experiments were a prerequisite for the recovery experiments in **Figure 5 (previously Fig. 3)** to obtain insights into potential side-effects induced by double-treatments and clarify how strong the association between actin and microtubules at the poles after nocodazole treatment is.

Figure B | Impaired membrane integrity after short exposure to 5 μ M latrunculin-B. Immunofluorescence images showing examples of MII oocytes treated with 5 μ M latrunculin-B for 5 or 10 minutes and stained for DNA, F-actin, and α -tubulin. Scale bar, 20 μ m. Projections from 3 sections of 1 μ m.

In summary, the separate and subsequent administration of the drugs helped us to assess potential competing or synergistic effects between the two cytoskeletal systems on spindle dynamics. This has been now mentioned in the manuscript.

...“Additionally, we treated oocytes with latrunculin-B prior to nocodazole addition to clarify if the disturbance of both cytoskeletal structures would have additional consequences on spindle architecture and chromosome clustering, which was not the case (Supplementary Fig. 6c).”...

...“The subsequent use of destabilizing and stabilizing agents was a helpful tool to better visualize the strongly compromised cytoskeletal structures at the spindle poles after microtubule impairment and to validate our previous finding that the interplay between the two cytoskeletal systems is sensitive to the polymerization state of microtubules. Both procedures yielded large multipolar spindles (Fig. 3h, Videos 8, 9); however, a communication between actin and microtubules was only observed in monastrol treated oocytes, where actin maintained its clear association with the spindle poles. This was reflected in the pronounced accumulation of filamentous actin structures radially spreading from the poles into the spindle interior (Fig. 3h, Video 10).”...

3. Without careful controls and more detailed characterization I am not sure that the anti-actin and anti-tropomyosin antibody stainings are particularly informative, as these antibodies are known for their tendency to stain off targets. In my opinion phalloidin and Lifeact already provides sufficient evidence for the presence of the observed F-actin structures.

We tested the anti-tropomyosin and anti-actin antibodies in different cell types to confirm their specificity (**Figures C, D**). Additionally, we would like to refer to the publications, which have been included to the manuscript: a) *Dugina V et al., (2009) J Cell Sci.; 122(Pt 16):2980-2988*; b) *Latham SL et al. (2018) Nat Commun., 9(1):4250*; c) *Pathan-Chhatbar S. et al., (2018) J. Biol. Chem. 293(3):863-875*.

[Redacted]

[Redacted]

We kept the experiments with anti-actin and anti-tropomyosin, since they provide information about the pool of actin present in and around the spindle, which has so far not been visualized in human oocytes and offers perspectives for future research addressing regulatory aspects of actin at the spindle.

Reviewer #2 (Remarks to the Author):

Comments to the authors

The authors report that the existence of actin spindles during meiosis I and II in human oocytes. This first becomes evident in late meiosis I and the actin fibres appear between the two masses of chromatin during anaphase I. Microtubule spindle reassembly at meiosis II appears to be accompanied by reassembly of the actin spindle. Using drug treatments to disrupt the microtubule and actin spindles, the authors conclude that actin spindle dynamics follow those of the microtubule spindle. The finding that actin infiltrates the spindle in human oocytes is consistent with a previous report, which included mouse and human oocytes (Mogessie and Schuh, 2017, Science). The current manuscript builds on this work by further characterising the behaviour of actin fibres during progression through meiosis in human oocytes.

Major comments:

The study is based on oocytes that were either not suitable for use in ICSI treatment, or failed to undergo fertilisation. A major caveat, not mentioned by the authors, is that these oocytes may be inherently abnormal and may therefore not provide an accurate picture of how healthy human oocytes progress through the meiotic divisions.

The reviewer mentions an important point. Unfortunately, we cannot warrant good or constant quality of oocytes, which is a general problem of studies with human oocytes or other human material. One criterion of donation was that couples were assigned to ICSI treatments due to male factor infertility, meaning that the oocytes were not per se inherently unhealthy. Of course, oocytes with signs of perturbation like degeneration and other morphological features indicative of impaired quality (visible by means of light-microscopy) were excluded. The **Figure E** below shows a staining example of such a degenerated oocyte. It is important to mention, that the study included oocytes derived from a heterogeneous group of women regarding health, age, lifestyle, and intensity of hormonal stimulation, all of which contribute to oocyte quality. This aspect has been added to the Methods section. Nonetheless, each of the various experiments yielded consistent results that taken together add to a conclusive model of interdependence between the two cytoskeletal systems during meiosis, regardless of whether individual oocytes were healthy or not at the time of analysis.

Figure E | Degenerated oocyte. Immunofluorescence projections of cell center (left) and spindle area (right) of a degenerated oocyte stained for chromosomes, actin, α -tubulin and γ -tubulin. $z = 11$ and 4 sections. Scale bars, 20 μm .

In addition, the authors make no effort to explain the provenance of the oocytes used in each experiment. This makes it very difficult to judge the quality and reliability of the data. For example, it is not at all clear what criteria were used to assign meiotic stages to the fixed oocytes shown in Fig 1. Were the oocytes matured in vitro and fixed at specific time points? Or were the stages assigned based on the configuration of the chromosomes and spindle. If the latter, how can one distinguish between a prometaphase oocyte and one that simply failed to assemble a spindle, which would not be unexpected for this source of oocytes. It would therefore be important to clarify the status of oocytes used for each experiment, if possible providing the time from administration of the hCG injection used to trigger maturation in vivo.

According to the procedure of oocyte retrieval and preparation, the status of each oocyte could be assigned. The meiotic stages illustrated in **Figure 1d** were obtained with oocytes that were in an immature germinal vesicle stage at the time of surgical retrieval and subsequently underwent maturation till they were fixed. These oocytes received no ICSI treatment and matured in vitro for 20 to 26 hours. As shown by Holubcova and Schuh, 2015, the maturation of human oocytes from the GV stage to the MII stage takes about 25 hours. Since we allowed the oocytes to initiate their maturation process spontaneously and independent of a pharmacological trigger, we obtained different maturation stages.

The meiotic stages have been assigned according to *i*) the absence or presence of a polar body, *ii*) configuration of the chromosomes, *iii*) spindle morphology and spindle position. We have added an additional panel in **Figure 2b** showing the whole dimension of the oocytes that helps in discriminating the stages. The prometaphase II stage can be clearly assigned, because *i*) of the presence of a polar body, *ii*) extruded chromosomes into the polar body *iii*) 2nd set of chromosomes not aligned at the spindle, *iv*) microtubules still projecting into the polar body (different to later stages), and *v*) the spindle has yet not acquired the typical MII architecture (**Figure F**). The morphology is indicative of a stage of spindle restructuring to be placed between the telophase I and metaphase II. We included the following sentences for clarity:

...” The status of each oocyte could be assigned on the basis of i) the presence or absence of a polar body to distinguish between early and later stages of meiosis and ii) spindle architecture and chromosome configuration, both adopting characteristic morphologies in prophase, metaphase, and anaphase stages (Fig. 2a).”...

Figure F | Prometaphase II in human oocytes. Immunofluorescence image depicting the characteristics used to identify prometaphase II in fixed oocytes. Arrowhead marks microtubules connecting oocyte and polar body. $z = 11$ section of $0.5 \mu\text{m}$. Scale bar, $20 \mu\text{m}$.

In the case of prometaphase I, the referee is right that a clear assignment is not possible. However, the configuration of chromosomes and microtubules in the stage assigned as prometaphase I, resembles strongly that previously described in Holubcová, Schuh, 2015. To avoid any confusion, we removed this panel. All other experiments were performed with MII oocytes that received ICSI treatment but had not formed pronuclei at the time of donation (24 hours after sperm injection). Importantly, the majority of MII oocytes retrieved by this procedure have formed a well-defined bipolar spindle with aligned chromosomes (see Figs. 1b, 3d). Drug addition experiments were initiated directly after donation and fixation was performed at specified time points indicated in the legends or methods section. Therein we describe all of the above aspects and also include information about the time hCG administration till start of experiments. The time window was about 57.5 to 61.5 hours.

...”

Collection and preparation of human oocytes

A total of 604 human oocytes were collected from 184 women receiving assisted reproductive treatment (ART) between October 2016 and May 2019. All oocytes donated for this study were retrieved to employ intracytoplasmic sperm injection (ICSI). Couples were assigned to ICSI treatment because of male factor infertility. Meiotic stages were investigated with oocytes that were in an immature germinal vesicle stage at the time of surgical retrieval and thus found unsuitable for sperm injection. These oocytes received no ICSI treatment and subsequently underwent spontaneous maturation in vitro for 20 to 26 hours independent of a pharmacological trigger till they were fixed. All other experiments were performed with MII oocytes that received ICSI treatment but had not formed pronuclei at the time of donation (24 hours after sperm injection). Only oocytes with no signs of perturbation, like degeneration or other morphological features indicative of poor quality by light microscopic inspection were included in the study independent of the condition of the donors in terms of health, age, lifestyle, and hormonal stimulation, all of which contribute to oocyte quality. Experiments were started immediately after donation, about 57.5 to 61.5 hours after hCG injection used to trigger maturation in vivo.”

...”

A further consideration in relation to the supply of oocytes, is that the ethics committee approval was granted for the use of unfertilised eggs (Methods section), yet it is clear from the data that some eggs contained sperm chromatin. Strictly speaking, these eggs are not unfertilized. Indeed, in some jurisdictions they would be legally considered as embryos. The authors, should therefore provide further clarification on the ethics approval to use eggs that contain sperm chromatin, but (presumably) do not form pronuclei.

These important points addressed by the referee have been included into the Methods chapter as required by the Journal. Donation and use of oocytes are in accordance with the ethics approval. For clarity, the present study includes only immature and ICSI treated oocytes that did not form pronuclei. The granted ethics approval goes beyond and allows experimental use of oocytes, which formed one and more pronuclei, however, without progressing through a pronuclear fusion 1 day after ICSI treatment. According to law, an oocyte is regarded as unfertilized as long as the pronuclei have not fused.

Minor comments:

The manuscript suggests that mouse oocytes at the metaphase II stage do not have a multi-polar spindle, but the cited article shows that this is not the case; though the number of foci in the example images from the cited paper and this manuscript suggest human metaphase II oocytes have more foci. Due to the nature of fixed data, it is surprising that the authors find their data support a case for multipolar intermediates in human oocyte spindle assembly; especially as Figure 1 suggests otherwise.

We thank the reviewer for the critical inspection of the images. We corrected our interpretation of the data as follows:

..." Given that the meiotic spindle in human oocytes progresses through highly instable multipolar and apolar stages till acquiring the typical bipolar configuration at metaphase II¹⁷, the constant association of actin with γ -tubulin at the spindle poles suggests that actin may use γ -tubulin centres to establish the communication with microtubules and thus participate in the assembly of a functional bipolar spindle. "...

It is surprising that culturing any cell containing a spindle for 25 minutes in 25 μ M nocodazole does not completely obliterate the microtubule spindle. Given the unexpected result, it would have been reassuring to see experiments in mouse oocytes using the same stock of nocodazole. Similarly, for the higher concentration of latrunculin B used to disrupt the actin cytoskeleton.

We regret that we cannot provide mouse data, since we currently do not have the resources and the authorized permission to perform animal experiments with super-ovulated mice for oocyte retrieval. However, we tested the stock of nocodazole using different concentrations and incubations time and also performed control experiments in HeLa cells. In human oocytes, longer incubation times (up to 60') and higher concentrations (up to 100 μ M, **see Supplementary Figure 6a below**) led to further reduction in spindle size, microtubule mass and disturbances in microtubule assembly but did not completely remove microtubules from the spindle, whereas nocodazole-addition to the HeLa cells (**Figure A**) obliterated all microtubules, demonstrating the effectiveness of the nocodazole stock used.

The increased stability of microtubules in human oocytes might be a specific feature of the human system responsible for the induction of spindle multipolarity as observed in all cases studied (**Fig. 3d in manuscript**). Please find below examples of nocodazole treated oocytes using different concentrations of the drug and varying incubation times.

With regard to the higher concentrations of latrunculin-B, we observed that spindle actin filaments were completely abolished and spindle morphologies notably altered already after short incubation times of 5-10 minutes, explaining the detection of residual solitary actin bundles within the cytoplasm.

During the course of our study, we tested various latrunculin concentrations and observed that short incubations (5 and 10 minutes) with 5 μM latrunculin not only depleted actin at the spindle but also drastically affected the entire oocyte (see Fig. B below) including partially disrupted cortical membranes and disturbed 3D-morphology.

Figure B | Impaired membrane integrity after short exposure to 5 μM latrunculin-B. Immunofluorescence images showing examples of MII oocytes treated with 5 μM latrunculin-B for 5 or 10 minutes and stained for DNA, F-actin, and α -tubulin. Scale bar, 20 μm . Projections from 3 sections of 1 μm .

We therefore used, e.g. for the recovery assay, lower concentrations of latrunculin, which disrupted spindle actin without destroying the oocyte. Importantly, our stock of latrunculin did effectively depolymerize actin filaments, as can also be seen in Fig. 4a in the manuscript. To avoid any confusion regarding the effectiveness of actin depolymerization we removed the actin channel from the corresponding images (Figure 4b, previous Figure 3a,b) and modified the main text as follows:

...“A complete depletion of microtubules was not achieved, even when higher concentrations of the drug were used and incubation times increased. This was surprising, because in mouse oocytes nocodazole is not known to induce spindle multipolarity^{14, 30}. Interestingly, longer incubations at higher doses increasingly reduced microtubule mass, affected chromosome alignment, and spindle shape (Supplementary Fig. 6a).”...

Supplementary Figure 6a | Effect of nocodazole on spindle microtubules in human oocytes. Immunofluorescence projections exemplifying the arrangement of chromosomes and microtubules in nocodazole-treated oocytes. $z = 11$ sections of 0.5 μm . Scale bars, 10 μm .

Figure A | Effect of nocodazole on microtubules in HeLa cells. Representative immunofluorescence images of HeLa cells treated with 5, 10, 25 μM nocodazole or 0.15% DMSO for 25 minutes showing DNA, α -tubulin and γ -tubulin. Data are from three independent experiments. $z = 5$ sections of $0.3 \mu\text{m}$. Scale bars, $10 \mu\text{m}$.

To reassess the effects of actin disruption, we performed additional experiments using different concentrations of cytochalasin-D. Notably, the effects were comparable with those previously reported in the mouse. $5 \mu\text{g/ml}$ cytochalasin-D (like in Mogessie and Schuh, 2017) also induced misalignment of chromosomes. The effect became more pronounced at $10 \mu\text{g/ml}$ with chromosomes distributing across misarranged microtubules. We included these data into **Figure 4 (previous Figure 3 in manuscript)**.

“...Actin assists microtubules in the assembly of a bipolar spindle in human oocyte meiosis
Next, we addressed whether actin disruption may also have an influence on microtubules. The addition of latrunculin-B abolished actin at the spindle with time but the spindle architecture remained unaffected only at low concentrations (Fig. 4a,d). Increased concentrations ($5 \mu\text{M}$) induced severe spindle disturbances within minutes. Some oocytes displayed a multipolar-like spindle geometry, in which microtubules were completely misarranged (Fig. 4b,d). To reassess the effects of actin depletion on spindle morphology and chromosome alignment, we chose an additional approach based on cytochalasin-D to disrupt actin. Strikingly, oocytes showed different types of chromosome configurations depending on drug concentration ranging from a few chromosomes being unaligned along the metaphase plate of still intact bipolar spindles to widely scattered chromosomes along severely misarranged microtubules (Fig. 4c,d). This is consistent with previous results obtained in the mouse system, where actin depletion in oocytes had drastic effects on chromosome alignment and clustering during metaphase II¹⁴. It is well known that the two drugs deplete actin by different mechanisms³¹. Yet the observed effects in human oocytes were largely similar with cytochalasin being more effective in disturbing the spindle architecture at low concentrations. “

Figure 4 | Disruption of filamentous actin affects chromosome alignment and perturbs bipolar spindle architecture. **a**, Representative z-projections of untreated human MII oocytes (upper panel) and oocytes treated with 2.5 μ M latrunculin-B (lower panel) showing chromosomes (Hoechst), actin (phalloidin), and microtubules (α -tubulin); $z = 16$ and 11 sections. **b**, Examples of MII oocytes treated with 1 μ M (low-dose) or 5 μ M (high dose) latrunculin-B showing chromosomes (Hoechst), actin (phalloidin), and microtubules (α -tubulin); $z = 4$ sections. **c**, Examples of MII oocytes treated with 5 μ g/ml (low-dose) or 10 μ g/ml (high-dose) cytochalasin-D showing chromosomes (Hoechst), actin (phalloidin), and microtubules (α -tubulin); $z = 4$ sections.

In addition, the authors should also consider the possibility of artefacts due to the addition of paclitaxel in the fixative. This would be expected to have a similar effect to taxol treatment (though the concentration is considerably lower).

We have considered this aspect and performed additional control experiments without adding paclitaxel to the fixative solution. There is no difference with regard to spindle size or spindle morphology between the oocytes fixed with taxol and those without, as shown in two representative figures for oocytes in telophase I and metaphase II (**Figure G below**).

[redacted]

The implication in the methods section is that MI oocytes were injected with sperm for ICSI treatment. This would not be considered normal practice.

It is indeed not the regular practice. However, when patients have only a very small number of ovarian follicles after hormonal treatment, the clinicians also take premature MI oocytes for ICSI to increase the probability of blastocyst formation. Since successful fertilization of MI oocytes is very rare, we found them not suitable for analysis. We removed the sentence from the methods section, which was included by mistake.

In addition, the authors comment on a second spindle forming around the paternal genome. Presumably, this observation was made in eggs that failed to form pronuclei 24 hours after sperm injection. If so, the authors should mention that it may not be reflective of normal progression thought fertilisation.

Exactly, pronuclei had not been formed in these ICSI-treated oocytes after 24 hours. The second spindle around the paternal genome is a side effect of ICSI and we agree that this is not representative of a normal fertilization. We used this example to visualize how close actin associates with microtubules, even within the spindle-like microtubule formation around the paternal genome. We describe this aspect in more detail in the main text:

“...We found a similar organization of actin inside the spindle of diploid oocytes (Supplementary Fig. 1a) and surprisingly, actin was also present in a bipolar barrel-shaped configuration within the spindle-like microtubule assembly formed around the paternal genome, a side side-effect of intracytoplasmic sperm injection (Supplementary Fig. 1b), which demonstrates that the communication of the two cytoskeletal systems is not restricted to microtubules associating with the maternal DNA.”...

A final comment would be that it seems unusual that the authors used 9% O₂ to culture in media that is typically used at either atmospheric or 5% O₂.

The use of 9% O₂ is mainly based on the experience of the laboratory staff at the Clinics and influenced by trends, which suspected atmospheric O₂ concentrations (21%) to be less suitable for ex-vivo oocyte cultivation due to the formation of oxygen radicals. For the medium itself, there are no recommendations regarding the O₂ as long as the CO₂ is held at 6%.

The title of the manuscript seems overblown: the authors do not provide evidence to support the statement that actin drives meiosis in human oocytes.

We agree with the referee and changed the title as follows:

“Actin-microtubule interplay coordinates spindle assembly in human oocytes.”

Similarly, I am not convinced that the following sentence in the concluding paragraph is supported by the evidence presented in the manuscript - “the herein observed tight interplay between actin and spindle microtubules provides a mechanistic basis for how actin accompanies the process of spindle migration, anchorage, and chromosome segregation for successful oocyte maturation”

We removed the corresponding sentence in the revised version and added a new paragraph to the discussion as follows:

...“ Age-related changes in microtubule acetylation have been reported to directly affect spindle function^{37, 38}. In light of the critical role of posttranslational modifications on microtubule stability, similar mechanisms altering actin dynamics including the regulation by tropomyosin copolymers may also affect spindle stability with implications for fertility and embryo development. In summary, our findings open new perspectives for future investigations addressing the signals and regulatory mechanisms underlying the crosstalk between actin and microtubules during meiosis and subsequent stages of embryo development that involves the formation of dual-spindles during transition from meiosis to mitosis³⁹. ” ...

In summary, the authors deserve credit for studying such a large number of human oocytes, and for the very nice imaging. However, as discussed above, the value to the scientific community will be limited without further information on the oocytes used in each experiment.

We thank the referee for appreciating our work and hope that the new version of our manuscript addresses the concerns to her/his satisfaction.

Reviewer #3 (Remarks to the Author):

This manuscript describes the interplay between actin and microtubules in the meiotic spindle in human oocytes. First, the authors confirmed that actin forms a spindle-like structure tightly associated with spindle microtubules in human oocytes, as previously observed (Mogessie and Schuh, Science 2017). They then further investigated the details of the actin spindle, providing some new observations of actin distributions during anaphase I and telophase I, and identifying tropomyosin and actin isoforms as components of the actin spindle. To address whether microtubules regulate the actin spindle, they treated human oocytes with drugs and conditions that affect microtubule dynamics. These experiments showed that the structural integrity of the actin spindle depends on microtubules, consistent with previous observations in mouse oocytes (Mogessie and Schuh, Science 2017). Finally, they addressed whether actin regulates the microtubule-based spindle, by perturbing actin dynamics during spindle reformation after nocodazole washout. The results appeared to suggest the importance of actin for microtubule-based spindle organization in human oocytes, which is again in agreement with the previous findings that actin contributes to spindle microtubule dynamics in mouse oocytes (Mogessie and Schuh, Science 2017).

Overall, the manuscript well describes the actin spindle and its functional relationships with microtubules in human oocytes. The observations presented in this manuscript are largely in line with what have been reported in mouse oocytes, therefore novelty is limited here. However considering that significant differences between mouse and human oocytes have been reported, datasets from studies using human oocytes are very important for the understanding of egg aneuploidy, the leading cause of pregnancy loss and several congenital disease. The image dataset and analysis of human oocytes presented here are of high quality, and therefore precious and likely attract readers from the fields of meiosis and reproductive biology and medicine. I would support publication of the manuscript in Nature Communications if the authors appropriately address my comment below.

Major comment:

The number of oocytes tested in the experiment shown in Figure 3 is too small to make any clear conclusions. The authors should repeat the experiment to have a number of oocyte sufficient to test a significant difference between the two groups. This is critical because this experiment addresses the functional dependency of microtubules on actin in this manuscript. Conclusive data should be provided to support their claim of 'Actin-microtubule interplay'.

Despite the restricted availability of human oocytes, we were able to repeat the recovery experiments increasing the n-values from previously 9 and 8 to 20 and 18, which allowed us to test for significance. These data have now been included into **Figure 5 (previously Figure 3)**. We have to admit that in the first version of the manuscript we made a mistake regarding the number of recovered versus not recovered oocytes depicted. Not 3 but 2 oocytes out of 9 failed to recover. The higher n-values are in line with the previous results and support further the functional dependence of microtubules on actin. Additional experiments with cytochalasin-D further discussed below are in agreement with the mouse data by Mogessie and Schuh, 2017, and provide conclusive evidence for a mutual dependence between the two cytoskeletal systems in meiosis.

Figure 5 | Actin supports microtubules in the assembly of a bipolar spindle. **a**, Proportion of oocytes that restored bipolar spindle morphology (‘recovered’) in the absence or presence of 1 μ M latrunculin-B after nocodazole treatment. **b**, Representative z-projections of recovered MII oocytes after nocodazole treatment and 60 to 90 minutes regeneration time in the absence or presence of 1 μ M latrunculin-B showing chromosomes (Hoechst), α -tubulin, and actin (phalloidin); $z = 4$ sections. **c**, Proportion of oocytes that retained spindle multipolarity (‘not recovered’) in the absence or presence of 1 μ M latrunculin-B after nocodazole treatment. **d**, Representative z-projections of not recovered MII oocytes after nocodazole treatment and 60 to 90 minutes regeneration time in the absence or presence of 1 μ M latrunculin-B showing chromosomes (Hoechst), α -tubulin, and actin (phalloidin); $z = 4$ and 5 sections. Scale bars, 10 μ m. n = total number of oocytes. The number of donors is specified within parentheses. Significance was tested using Fisher’s exact test. Error bars represent standard error of the mean.

To convince the contribution of actin to the microtubule spindle, it is recommended that authors add an experiment using another perturbation of actin dynamics. Currently the manuscript uses only Latrunculin, suggesting that spindle reformation is Latrunculin-sensitive. This may imply that actin contributes to spindle formation, but a concern here is a secondary effect of Latrunculin. If consistent results could be obtained by another perturbation (e.g. Cyto B or D, other inhibitors or RNAi?), their claim of 'Actin-microtubule interplay' would be further supported.

We followed the advice of the referee and performed additional experiments using different concentrations of cytochalasin-D to reassure the effects of actin disruption. These data are included in **Figure 4 (previous Figure 3)**. Notably, the effects were comparable with those previously reported for mouse oocytes. We observed at 5 µg/ml cytochalasin-D (like in Mogessie and Schuh, 2017) partially misaligned chromosomes. The effect became more pronounced in oocytes treated with 10 µg/ml cytochalasin-D, where chromosomes distributed widely separated along misarranged microtubules. Although it is well known that the two drugs deplete actin by different mechanisms, the effects in the oocytes are comparable further supporting the importance of a molecular interplay between microtubules and actin in spindle assembly. These data also lend support to the view of a conserved role of actin in assisting the process of chromosome alignment in mammalian oocytes, including human oocytes.

*“...Actin assists microtubules in the assembly of a bipolar spindle in human oocyte meiosis
Next, we addressed whether actin disruption may also have an influence on microtubules. The addition of latrunculin-B abolished actin at the spindle with time but the spindle architecture remained unaffected only at low concentrations (Fig. 4a,d). Increased concentrations (5 µM) induced severe spindle disturbances within minutes. Some oocytes displayed a multipolar-like spindle geometry, in which microtubules were completely misarranged (Fig. 4b,d). To reassure the effects of actin depletion on spindle morphology and chromosome alignment, we chose an additional approach based on cytochalasin-D to disrupt actin. Strikingly, oocytes showed different types of chromosome configurations depending on drug concentration ranging from a few chromosomes being unaligned along on the metaphase plate of still intact bipolar spindles to widely scattered chromosomes along severely misarranged microtubules (Fig. 4c,d). This is consistent with previous results obtained in the mouse system, where actin depletion in oocytes had drastic effects on chromosome alignment and clustering during metaphase II¹⁴. It is well known that the two drugs deplete actin by different mechanisms³¹. Yet the observed effects in human oocytes were largely similar with cytochalasin being more effective in disturbing the spindle architecture at low concentrations. “*

Figure 4 | Disruption of filamentous actin affects chromosome alignment and perturbs bipolar spindle architecture. **a**, Representative z-projections of untreated human MII oocytes (upper panel) and oocytes treated with 2.5 μ M latrunculin-B (lower panel) showing chromosomes (Hoechst), actin (phalloidin), and microtubules (α -tubulin); $z = 16$ and 11 sections. **b**, Examples of MII oocytes treated with 1 μ M (low-dose) or 5 μ M (high dose) latrunculin-B showing chromosomes (Hoechst), actin (phalloidin), and microtubules (α -tubulin); $z = 4$ sections.

Reviewers' comments:

Reviewer #2 (Remarks to the Author):

The authors have done a considerable amount of work to address the reviewers' comments. There are however, some outstanding issues (below), that should be addressed. Otherwise, the manuscript is, in my view, suitable for publication. I congratulate that authors on adding a substantial contribution to the body of knowledge on spindle dynamics in human oocytes.

Response to reviewer 2 comments

1. In the interest of the readers' ability to interpret this work and to see where it fits in the context of other studies on human oocytes, the authors should state explicitly that the oocytes they are using failed to resume meiosis following ovarian stimulation. This should be made clear in the Introduction and Results as well as in the Methods section. In asking this, I would like to stress that I am not trying to detract from this nice study. However, I firmly believe that, in the interest of those who come behind us, it behoves us as researchers to describe our experimental systems as clearly and as comprehensively as possible.

2. In addition, in the Methods section, the authors state that the egg donors were undergoing ICSI treatment because of male infertility. This does not preclude the existence of female factors and should be clarified.

Additional points

1. Abstract - last line. The term "genetic material" would also include the mitochondrial DNA. To avoid ambiguity, the authors should specify that they refer here to the nuclear genome.

2. Line 18: "unstable" rather than "instable"

3. Supplementary Fig 1a and line 20: What do the authors mean by diploid MII oocytes? This seems contradictory and is bound to cause confusion.

4. Line 25-26: These sentences give the impression that humans are not mammals.

Reviewer #3 (Remarks to the Author):

In the revised manuscript, the authors partly addressed my comments.

One of my concerns was the too small number of oocytes in the experiment of nocodazole-washout with Latrunculin (previously in Figure 3). They now show the data of sufficient numbers of oocytes and the results of statistical tests (now in Figure 5). This supports their conclusion.

The other my concern was a secondary effect of Latrunculin. To address this they now include a result of cytochalasin treatment in Figure 4. In the text, they claim that a high dose of cytochalasin disrupts spindle organization. However this is difficult to judge from the presented dataset, which lacks information about how many oocytes were observed and how many showed disorganized spindles. The dataset would be convincing only if they could show a statistical significance in the dose-dependent effects of cytochalasin.

As I previously noted, whether actin functionally contributes to microtubule organization or not is a critical part of this study for claiming actin-microtubule interplay. Evidence for actin contribution should be robustly provided.

We thank the reviewers for the new revision of our manuscript and their valuable comments. We have addressed the points raised by the referees accordingly. All major modifications made to the manuscript are highlighted in the manuscript file.

Reviewers' comments:

Reviewer #2 (Remarks to the Author):

The authors have done a considerable amount of work to address the reviewers' comments. There are however, some outstanding issues (below), that should be addressed. Otherwise, the manuscript is, in my view, suitable for publication. I congratulate that authors on adding a substantial contribution to the body of knowledge on spindle dynamics in human oocytes.

We are grateful for this comment and thank the reviewer for the credit.

1. In the interest of the readers' ability to interpret this work and to see where it fits in the context of other studies on human oocytes, the authors should state explicitly that the oocytes they are using failed to resume meiosis following ovarian stimulation. This should be made clear in the Introduction and Results as well as in the Methods section. In asking this, I would like to stress that I am not trying to detract from this nice study. However, I firmly believe that, in the interest of those who come behind us, it behoves us as researchers to describe our experimental systems as clearly and as comprehensively as possible.

We have included this aspect in the revised version of the manuscript as highlighted in:

Results section:

"...We used oocytes that were immature at the time of surgical retrieval following ovarian stimulation. The maturation of human oocytes from prophase I till the stage of MII arrest takes approximately 25 hours¹⁷. We therefore allowed oocytes to resume maturation independent of any pharmacological trigger in vitro yielding oocytes with different maturation stages. The status of each oocyte could be..."

Methods section:

"...Couples were assigned to ICSI treatment because of male factor infertility. Meiotic stages were investigated with oocytes that failed to resume meiosis following ovarian stimulation and were in an immature germinal vesicle stage at the time of surgical retrieval, thus unsuitable for sperm injection."...

2. In addition, in the Methods section, the authors state that the egg donors were undergoing ICSI treatment because of male infertility. This does not preclude the existence of female factors and should be clarified.

"Only oocytes with no signs of perturbation, like degeneration or other morphological features indicative of poor quality by light microscopic inspection were included in the study independent of the condition of the donors in terms of health, age, lifestyle, and hormonal stimulation, all of which contribute to oocyte quality. Therefore female factors affecting oocyte quality cannot be excluded. Experiments were started immediately after donation, about 57.5 to 61.5 hours after hCG injection used to trigger maturation in vivo."

Additional points

1. Abstract - last line. The term “genetic material” would also include the mitochondrial DNA. To avoid ambiguity, the authors should specify that they refer here to the nuclear genome.

Has been modified.

2. Line 18: “unstable” rather than “instable”

Has been modified.

3. Supplementary Fig 1a and line 20: What do the authors mean by diploid MII oocytes? This seems contradictory and is bound to cause confusion.

We clarified this aspect as highlighted:

*...“**Supplementary Figure 1** | Actin pervades the spindle of an oocytes with abnormal chromosome number and associates with microtubules surrounding spermal DNA. a, Representative z-projection of a diploid metaphase II oocyte stained for chromosomes (Hoechst) and actin (phalloidin); z = 25 sections. b, Immunofluorescence staining of γ tubulin together with chromosomes (Hoechst) and actin (phalloidin) at a spindle structures surrounding the spermal DNA in an ICSI treated oocyte. Asterisk marks spermal centrosome; z = 13 sections. Scale bars, 10 μ m.”...*

In the main text:

*...“We found a similar **three-dimensional** organization of actin inside the spindle of **developmentally disturbed oocytes that failed to extrude redundant chromosomes in meiosis I and still contained the two sets of nuclear genetic material in meiosis II (Supplementary Fig. 1a).**”....*

4. Line 25-26: These sentences give the impression that humans are not mammals.

This aspect has been clarified as follows:

*...“**In oocytes of many species**, the formation of a bipolar spindle proceeds through the self-organization of acentriolar microtubule organizing centres (MTOCs), which replace the function of centrosomes. Human oocytes are in this respect different **since they lack typical MTOCs**^{17, 19}; however, they are still capable to configure a bipolar spindle and gather microtubules at the poles.”....*

Reviewer #3 (Remarks to the Author):

In the revised manuscript, the authors partly addressed my comments.

One of my concerns was the too small number of oocytes in the experiment of nocodazole-washout with Latrunculin (previously in Figure 3). They now show the data of sufficient numbers of oocytes and the results of statistical tests (now in Figure 5). This supports their conclusion. The other my concern was a secondary effect of Latrunculin. To address this they now include a result of cytochalasin treatment in Figure 4. In the text, they claim that a high dose of cytochalasin disrupts spindle organization. However this is difficult to judge from the presented dataset, which lacks information about how many oocytes were observed and how many showed disorganized spindles. The dataset would be convincing only if they could show a statistical significance in the dose-dependent effects of cytochalasin.

We appreciate the comment of reviewer #3 and performed the analysis accordingly. We agree with the reviewer that actin depletion experiments do not allow clear assignments of dose-dependent effects. We therefore quantified the changes in chromosome alignment induced by actin depletion. We believe that dose-dependent experiments do not add considerably new information than the quantification data in Figure 4e already provide, an estimate of how actin depletion affects chromosomes. Since the reported data summarized in Fig.5 are based on latrunculin, we would like to emphasize the results obtained with latrunculin, for which we observe significant differences. These new data have now been included as additional panel in Figure 4 and in the main text.

Main text:

....“This is consistent with previous results, where actin depletion had drastic effects on chromosome alignment during metaphase II in mouse oocytes¹⁴. Careful inspection of the untreated and latrunculin-treated oocytes allowed us to estimate how severely actin depletion affects chromosome alignment. We quantified the portion of oocytes containing misaligned chromosomes, i.e. more than two chromosome localizing outside the metaphase plate or scattering around the poles. Chromosome disarrangements could result from weakened chromosome attachments or spindle instabilities including loss of spindle bipolarity^{14, 17}. In the untreated group, the majority of oocytes (12/18) displayed well aligned chromosomes and normal spindle architectures. Notably, the proportion of intact chromosome arrangements was reduced, when oocytes were treated with latrunculin (5/17) or exposed to cytochalasin (2/6) (Fig. 4d). These data agree with previous findings in mouse oocytes that actin is essential for chromosome alignment providing further evidence for a conserved role of actin at the meiotic spindle throughout mammalian species³².”.....

Figure 4 | Disruption of filamentous actin affects chromosome alignment and perturbs bipolar spindle architecture. *a*, Representative z -projections of untreated human MII oocytes (upper panel) and oocytes treated with $2.5 \mu\text{M}$ latrunculin-B (lower panel) showing chromosomes (Hoechst), actin (phalloidin), and microtubules (α -tubulin); $z = 16$ and 11 sections. *b*, Examples of MII oocytes treated with $1 \mu\text{M}$ (upper panel) or $5 \mu\text{M}$ (lower panel) latrunculin-B showing chromosomes (Hoechst), actin (phalloidin), and microtubules (α -tubulin); $z = 4$ sections. *c*, Examples of MII oocytes treated with $5 \mu\text{g/ml}$ (upper panel) or $10 \mu\text{g/ml}$ (lower panel) cytochalasin-D showing chromosomes (Hoechst), actin (phalloidin), and microtubules (α -tubulin); $z = 4$ sections. *d*, Comparison of oocytes with aligned vs unaligned chromosomes in the absence or presence of actin depolymerizing agents ($1, 2.5, 5 \mu\text{M}$ latrunculin-B incubations for 5 to 60 minutes prior to fixation; 5 mg/ml and 10 mg/ml cytochalasin-D 25 and 60 minutes prior to fixation). Number of donors is specified within parentheses. Significance was tested using Fisher's exact test. Error bars represent standard error of the mean. *e*, Schematic illustration of chromosomes, microtubules and actin at the spindle of MII oocytes upon treatment with actin depolymerizing agents.

As I previously noted, whether actin functionally contributes to microtubule organization or not is a critical part of this study for claiming actin-microtubule interplay. Evidence for actin contribution should be robustly provided.

We hope the additional data provide further information supporting our claim.

REVIEWERS' COMMENTS:

Reviewer #3 (Remarks to the Author):

My concerns have been satisfactorily addressed. I appreciate this nice work.